# Coherent long-distance displacement of individual electron spins

H. Flentje[1,2], P.-A. Mortemousque[1,2], R. Thalineau[1,2], A. Ludwig [3], A.D. Wieck[3], C. Bäuerle [1,2] & T. Meunier[1,2]

Controlling nanocircuits at the single electron spin level is a possible route for large-scale quantum information processing. In this context, individual electron spins have been identified as versatile quantum information carriers to interconnect different nodes of a spin-based semiconductor quantum circuit. Despite extensive experimental efforts to control the electron displacement over long distances, maintaining electron spin coherence after transfer remained elusive up to now. Here we demonstrate that individual electron spins can be displaced coherently over a distance of 5 μm. This displacement is realized on a closed path made of three tunnel-coupled lateral quantum dots at a speed approaching 100 ms$^{-1}$. We find that the spin coherence length is eight times longer than expected from the electron spin coherence without displacement, pointing at a process similar to motional narrowing observed in nuclear magnetic resonance experiments. The demonstrated coherent displacement will open the route towards long-range interaction between distant spin qubits.

[1] University of Grenoble Alpes, Institut NEEL, F-38042 Grenoble, France. [2] CNRS, Institut NEEL, F-38042 Grenoble, France. [3] Lehrstuhl für Angewandte Festkörperphysik, Ruhr-Universität Bochum, Universitätsstraße 150, D-44780 Bochum, Germany. Correspondence and requests for materials should be addressed to T.M. (email: tristan.meunier@neel.cnrs.fr)

While it is clear by now that the spin degree of freedom of an electron is an interesting building block for processing and storing quantum information[1–6], important questions concerning the system scalability remain to be addressed before building a large-scale spin-based quantum processor. Ultimately, the problem reduces to the ability to transfer quantum information on a chip. Following the work on superconducting qubits, significant experimental efforts are currently focusing on the possibility to couple distant electron spins via a quantum mediator[7–9]. An alternative way consists in displacing the electron spin itself[2]. One possibility is to convey the electron in moving quantum dots defined by surface acoustic waves, where it is trapped and propagates isolated from the surrounding electrons at the speed of sound[10–13]. Even though electron and spin transfer have been demonstrated, the technology of moving quantum dots at the single electron level is not yet controlled well enough to investigate coherence properties[12]. A more conventional strategy consists in displacing the electron in an array of tunnel-coupled quantum dots[14–22]. So far, only a classical spin shuttle over linear arrays of three and four dots has been demonstrated, whereas slow electron displacement on a closed loop has been demonstrated in a four-quantum-dot system[17].

In this article, we demonstrate the coherent spin displacement of individual electrons in investigating the spin dynamics of two electrons initially prepared in a singlet spin state and displaced in an array of three lateral quantum dots defined in a circular geometry (a similar demonstration was recently realized in linear dot arrays[18]). When the electrons are displaced and explore a larger surface area than without motion, the main decoherence mechanism for static electron spins is averaged away and the coherence time of the singlet state is observed to be enhanced. Moreover, displacement-induced spin-flip processes are revealed with the dependence of the coherence time with the externally applied magnetic field and limit the distance over which electron spin coherence can be preserved.

## Results

**Electron displacement and spin measurement.** The triple-dot system is defined by Schottky gates with standard split-gate techniques in a two-dimensional electron gas (2DEG) residing at the interface of AlGaAs-GaAs heterostructure (see Fig. 1a and

"Methods" section). The two electrons are loaded into the system via the *bottom dot*. With large gate µs-movement, they are brought above the Fermi energy in the isolated configuration where the coupling to the electron reservoir can be ignored (see Supplementary Note 1)[23]. The resulting charge response of the electrometer when changing the chemical potentials of the three dots in the isolated configuration is presented in Fig. 1b. As expected, only six possible charge configurations are observed (see Supplementary Note 1 and Supplementary Fig. 1), rendering electron displacement in the circular triple dot system straightforward to implement. The tunnelling rates between the three dots are tunable up to the gigahertz regime. Nanosecond control of the gate voltages permits therefore adiabatic electron transfer between the dots faster than the spin coherence time[23].

The two-electron spin state after displacement can be inferred by bringing the electrons in the *bottom dot*, where exchange of electrons with the reservoir is possible[23]. In the two-electron case, the ground singlet (S) and the three excited triplet ($T_+$, $T_0$, $T_-$) states are distinguished using the tunnel-rate spin read-out method with a single-shot fidelity of 80% (see Supplementary Note 2 and Supplementary Fig. 2)[24, 25]. Unless specified, the experiments are repeated a thousand times to obtain the singlet probability.

**Electron spin coherence without displacement.** First, we specifically focus on the spin dynamics when the two electrons are static in two different dots. Initialization in the singlet ground state is performed by relaxation in the *bottom dot*. By rapidly pulsing the gate voltages, the electrons are separated into two different dots for a controlled duration $\tau_s$. In this way, we probe how long the phase coherence initially present in the singlet state can be preserved when the electrons are separated[16]. If the phase coherence is maintained, the system will remain in the singlet state. Otherwise, the final spin state will be a mixture of singlet and triplet states. Figure 2a presents where singlet-triplet spin mixing is occurring in the charge stability diagram. Each point is obtained by initialization to the singlet state at point R of Fig. 1b. Pulses of 50 ns duration and amplitude $V_1$ and $V_2$ are then simultaneously applied on the $V_{M,1}$ and $V_{M,2}$ gates, respectively. The resulting two-electron spin state population for each ($V_1$, $V_2$) is then averaged over 150 single shot measurements. Three distinct regions where the spin mixing is efficient are observed. They correspond to the three charge configurations where the

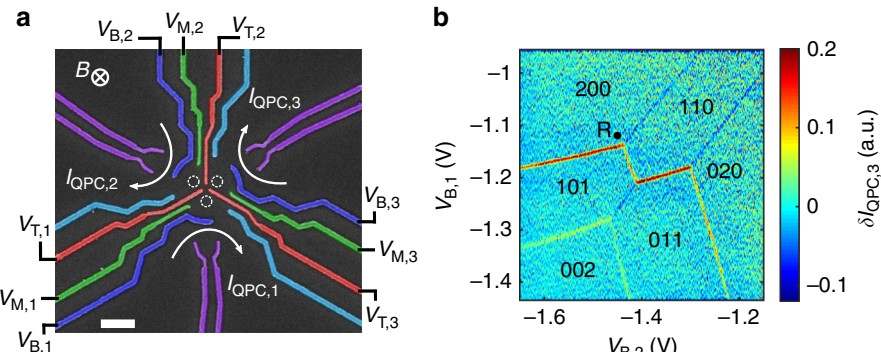

**Fig. 1** Triple dot system in a circular geometry. **a** Scanning electron microscope image of the circular triple dot sample. The position of the dots is shown in *white dashed circles*. The voltages applied on the *green* ($V_{M,1}$, $V_{M,2}$, $V_{M,3}$), *red* ($V_{T,1}$, $V_{T,2}$, $V_{T,3}$) and *blue gates* ($V_{B,1}$, $V_{B,2}$, $V_{B,3}$) allow to predominantly control the coupling between the dots, the coupling to the reservoirs and the dot-chemical potentials, respectively. The *purple gates* are used to define sensing dots to probe, with $I_{QPC,1}$, $I_{QPC,2}$ and $I_{QPC,3}$, the charge configuration of the triple-dot system. Electron loading and spin read-out are realized in the *bottom dot*. A magnetic field B is applied perpendicular to the sample. *Scale bar* is 300 nm. **b** Derivative $\delta I_{QPC,3}$ of $I_{QPC,3}$ along $V_{B,1}$ when the system is scanned in the two-electron isolated configuration with the gates $V_{B,1}$ and $V_{B,2}$. The label ($N_1$, $N_2$, $N_3$) corresponds to the number of electrons in the *bottom, top left* and *top right dots*, respectively

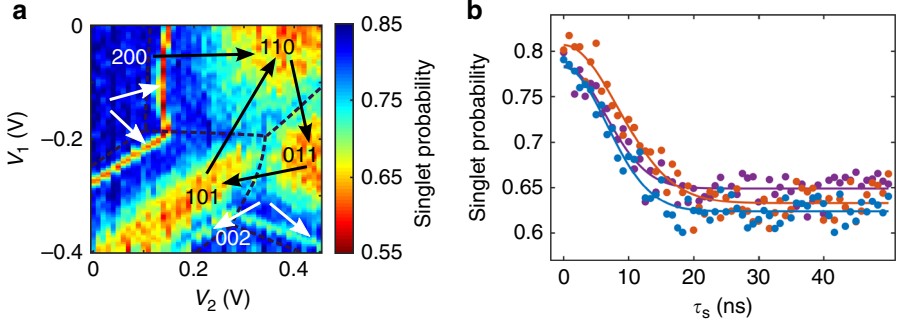

**Fig. 2** Spin-mixing map and coherence without displacement. **a** Two-electron spin mixing map at $B = 90$ mT. From the position R in Fig. 1b, 50-ns pulses of amplitude $V_1$ and $V_2$ are simultaneously applied on $V_{M,1}$ and $V_{M,2}$, respectively, before performing single-shot read-out of the two-electron spin states to extract the singlet probability. In the separated configurations, S mixes with $T_0$. In between two of these regions, one electron is exchanged between two dots and the spin mixing is less effective due to the increase of the exchange interaction (see Supplementary Note 3 and Supplementary Fig. 3). The *arrows* represent the ($V_1$, $V_2$) voltage pulses used to implement the separation and the displacement of the electrons and presented in Fig. 3a. The influence of the tunnel-coupling on the spin-mixing map is discussed in Supplementary Note 5. **b** Singlet probability as a function of the time $\tau_s$ spent in separated configurations, where the electrons are static in (0, 1, 1), (1, 1, 0) and (1, 0, 1), (*orange*, *blue* and *purple*, respectively). The data are fitted with a Gaussian decay with a characteristic time $T_2^*$, respectively, equal to $9.0 \pm 0.3$, $9.9 \pm 0.4$ and $10.2 \pm 0.4$ ns

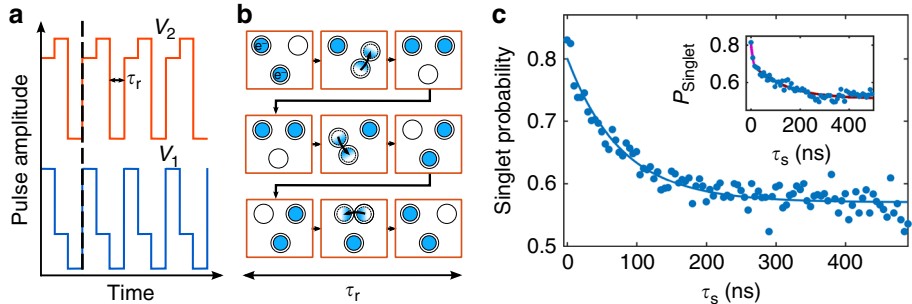

**Fig. 3** Coherent spin displacement. **a** Schematics of the time-dependent sequence applied on gates $V_1$ and $V_2$ to perform the electron displacement. $N_t$ is the number of loops performed by the electrons and $\tau_r$ is the duration spent in each charge configuration ((1, 1, 0), (0, 1, 1), (1, 0, 1)). Considering the rise time of the pulse generator, the electrons are adiabatically transferred between the dots (see "Methods" section). **b** Schematics of the spatial displacement of the electrons during a single rotation. Most of the displacement is occurring while the electrons are trapped in moving quantum dots. **c** Singlet probability as a function of the time $\tau_s$ for the case where the electrons are rotating between separated charge configurations with $\tau_r = 1.7$ ns (electron displacement with minimized static phase) and $B = 200$ mT. The data are fitted with an exponential decay with a characteristic time $\tau_{Decay}$ equal to 78 ns. *Inset*: Singlet probability as a function of the time $\tau_s$ for $B = 150$ mT and $\tau_r = 2.5$ ns. The data are fitted with a two-exponential decay. The *solid pink* (*red*) *line* is a fit of the fast (slow) decay. We notice very similar spin mixing time-evolution with displacement for different tunnel-coupling conditions (see Supplementary Note 6 and Supplementary Fig. 6)

electrons are separated in two dots. In these regions, the exchange interaction between the two electrons can be neglected and the system is dominated by the coupling to the nuclear spins of the heterostructure via hyperfine interaction[16, 23, 26]. At a magnetic field of 150 mT, the spin mixing occurs only between S and $T_0$ (see Fig. 2b). By varying $\tau_s$, we observe a Gaussian decay $e^{-\tau_s^2/T_2^{*2}}$ of the singlet probability characterized by a timescale $T_2^*$ close to 10 ns, very similar in each mixing region[16, 23]. In addition to the S-$T_0$ mixing regions, we notice four additional mixing lines, clearly separated from the other mixing regions. We attribute them to the mixing of S and $T_+$ states and their observation is a signature of large and coherent tunnel-coupling between the dots (see Supplementary Note 3)[23].

**Electron spin coherence with displacement**. We proceed to the investigation of the two-electron phase coherence while the two electrons are individually displaced on the closed loop formed by the three quantum dots. More specifically, the electrons are initially prepared in the singlet state of the (2, 0, 0) charge configuration. The system is then pulsed fast to the region (1, 1, 0),

where the electrons are separated in two dots and rotated repeatedly between the spin mixing regions of the (1, 1, 0), (0, 1, 1) and (1, 0, 1) charge configurations with voltage pulse sequence on $V_1$ and $V_2$ (see Figs. 2a and 3a). It leads to a series of quantum dot displacements and single electron tunnelling events schematically shown in Fig. 3b. Arbitrary long displacements can be implemented by repeating the loop. We control the number of loops $N_t$ performed by the electrons and the duration $\tau_r$ spent in each charge configuration. With displacement, the time $\tau_s$ where the electrons are separated and experience singlet-triplet mixing is equal to $3 N_t \tau_r$. Finally, the system is tuned back from the (1, 1, 0) to the (2, 0, 0) charge configuration, where spin read-out is performed. With $\tau_r$ equal to 1.7 ns, the resulting singlet probability is decaying as a function of $\tau_s$. These results demonstrate coherent electron spin transfer in an array of quantum dots in a circular geometry. It is worth noting that the decoherence law follows an exponential decay $e^{-\tau_s/\tau_{Decay}}$ with displacement (see Supplementary Note 4 and Supplementary Fig. 4). We fit a spin coherence time of 80 ns, almost eight times longer than for the static case at 200 mT (see Fig. 3c), only possible with a significant reduction of the influence of the hyperfine interaction during the electron displacement (for the

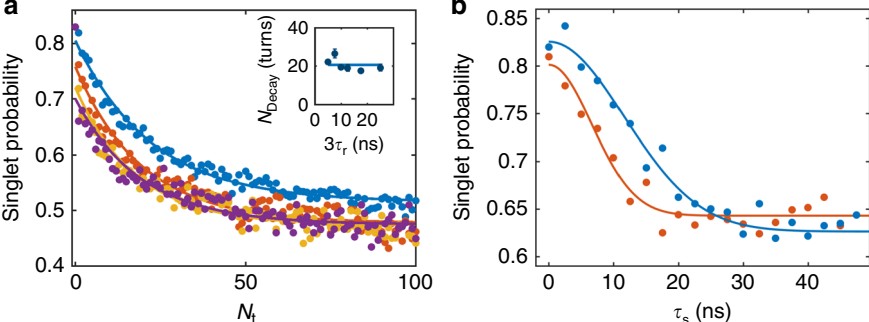

**Fig. 4** Influence of the number of turns on spin mixing and motional narrowing. **a** Singlet probability as a function of the number of turns $N_t$ for different time per turn $3\tau_r$ at $B = 90$ mT. The *blue, red, orange* and *purple curves* correspond, respectively, to 5, 10, 18 and 25 nanoseconds per turn. The fast decay associated to the static phase is observed for small number of turns. The data after the static phase are fitted with an exponential decay with a characteristic number of turns $N_{Decay}$. *Inset*: Extracted $N_{Decay}$ as a function of $3\tau_r$. The *solid line* is a constant fit to the data, which slightly depends on the gate voltage configuration (see Supplementary Note 6). **b** Singlet probability as a function $\tau_s$ for $N_t$ equal to one and obtained by increasing $\tau_r$ (electron displacement with minimized transfer phase) at $B = 90$ mT. The corresponding singlet probability as a function of $\tau_s$, where the electrons are static in the (0, 1, 1) charge configuration at 90 mT is plotted in *orange* for comparison. The data are fitted with a Gaussian decay with a characteristic time $T_2^*$ equal to $16.4 \pm 0.9$ ns ($9.4 \pm 0.4$ ns) for the *blue* (*orange*) curve. All the data presented in this figure are in a tunnelling condition slightly different than Figs. 2, 3 and 5 (see Supplementary Note 6 and Supplementary Fig. 6)

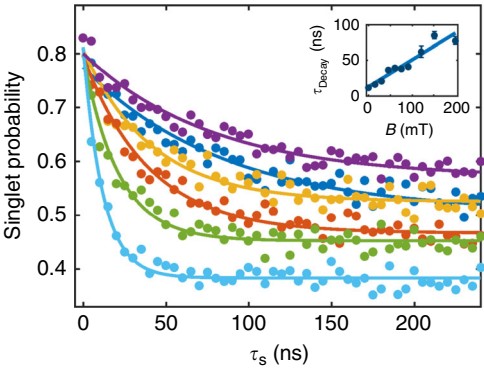

**Fig. 5** Influence of the magnetic field on the spin coherence time. Singlet probability as a function of the time $\tau_s$ for the case where the electrons are rotating in the triple-dot system for different $B$ at fixed $\tau_r = 1.7$ ns (electron displacement with minimized static phase). The *cyan, green, red, orange, blue* and *purple curves* correspond, respectively, to 0, 30, 61, 91, 152 and 200 mT. The data are fitted with an exponential decay with a characteristic time $\tau_{Decay}$. *Inset*: Extracted $\tau_{Decay}$ as a function of the magnetic field $B$. The *solid line* is a linear fit of the data

situation with zero magnetic field, see Supplementary Note 5 and Supplementary Fig. 5).

For $\tau_r$ set to 2.5 ns, the time dependence of the spin mixing no longer exhibits a single exponential behaviour but is characterized by two timescales (see *inset* in Fig. 3c). First, the system decays fast on a timescale similar to the coherence time in static dots, and then it evolves on a longer timescale towards a mixed singlet-triplet state. Figure 4a shows the spin singlet probability as a function of $N_t$ for different values of $\tau_r$. The long decay is only dependent on $N_t$, and the underlying decay mechanism must therefore depend dominantly on the number of tunnelling events or on the travelled distance of the electrons, rather than the separation time (the influence of the displacement geometry is discussed in Supplementary Note 7 and Supplementary Fig. 7). We interpret these observations as the consequence of two different phases during the displacement procedure: the static phase where the electrons are static in two different dots and the transfer phase where they are moving between two dots separated by ~110 nm on a fixed timescale corresponding to the rise time of the pulse generator (0.9 ns).

When the electrons realize only one rotation ($N_t = 1$) with increasing $\tau_r$ ($\tau_s = 3\tau_r$), the influence of the transfer phase is minimized. In this situation, the observed spin mixing time is only $1.74 \pm 0.17$ times longer than in the static configuration (see Fig. 4b). Additionally, the singlet probability decay is Gaussian hinting at a similar decoherence mechanism as in the case with no displacement.

For $\tau_r$ set to 1.7 ns, the time spent in the static phase is minimized and the spin decoherence process is mainly occurring during the transfer phase (see "Methods" section). In this case, the spin dynamics is characterized by a single exponential decay of the singlet probability (see Fig. 5). As the magnetic field is increased from 0 to 200 mT, we observe a progressive reduction of the singlet mixing with $T_+$ and $T_-$ after a 250 ns evolution and a linear increase of the spin coherence time, from 12 ns to almost 80 ns. At 200 mT, only mixing between S and $T_0$ is to a good approximation observed. Considering the estimated distance of 110 nm between the dots (see Supplementary Note 1), we measure a maximal spin coherence length of 5 μm.

## Discussion

The extremal cases of minimized static phase and minimized transfer phase allow us to address the important issue of the coherence limiting processes during the displacement. During the static phase, the electrons are experiencing a fast spin mixing similar to the case with no displacement. Moreover, the observed increase factor of the spin mixing time equal to 1.74 is very close to the $\sqrt{3}$-factor expected for an electron spin coupled to a three times larger nuclear spin bath[26, 27]. Therefore, these observations support a scenario where the main source of decoherence in the static phase is the longitudinal hyperfine interaction.

During the transfer phase, the electrons are displaced in moving quantum dots induced by the time-dependent potentials applied on the gates, before and after the tunnelling processes. As a consequence, the number of nuclei coupled to the electrons is drastically increased and a process similar to motional narrowing observed in liquid nuclear magnetic resonance experiments[27, 28] is expected to result in an increase of the electron spin coherence. This is in agreement with the observed decoherence law in Figs. 2b and 3c, which changes from Gaussian to exponential decays without and with displacement. Moreover, individual spin-flip processes, stimulated by the electron motion and resulting from

either spin-orbit or transverse hyperfine couplings, are then a possible decoherence channel for the electron spins[28, 29]. Due to the change of the dot position during the displacement procedure, both mechanisms are producing an effective time-dependent magnetic field $\Delta B$ on the displaced electrons leading eventually to a spin-flip. As a result, the two-electron system is able to leave the S-$T_0$ subspace (see Supplementary Note 5 and Supplementary Fig. 5). The smaller the change of the dot position, the smaller is $\Delta B$. The displacement procedure in moving quantum dots therefore implies that the amplitude of $\Delta B$ decreases with frequency. As only the $\Delta B$ component at the Larmor frequency leads to spin-flips, the spin-flip processes are expected to become less and less efficient as the magnetic field is increased from 0 to 200 mT. These considerations are in agreement with the experimental findings of an enhancement of the spin mixing time and a gradual increase of the singlet population after 250 ns evolution with magnetic fields presented in Fig. 5. At 200 mT, the system remains, to a good approximation, in the S-$T_0$ subspace and the spin coherence time is expected to be limited by the residual longitudinal hyperfine interaction after motional narrowing[28].

In conclusion, we have demonstrated that the coherence of a two-electron singlet state is preserved when the electrons are separated and displaced over 5 μm around a closed loop in a three-dot system. Compared to the situation without displacement, the spin coherence time is increased by a factor of eight via a motional narrowing process to 80 ns. Furthermore, spin-flip processes stimulated by the electron motion are found to limit the spin coherence time. The demonstrated coherent spin displacement could be a viable route to interconnect quantum nodes in spin-based quantum processors. On the more fundamental side, increasing the speed of the closed-loop transfer with larger tunnel-couplings should allow the exploration of non-abelian and holonomic spin manipulation[30–33] in future experiments.

## Methods

**Multi-dot system and experimental set-up.** The device is defined by Schottky gates in an n-$Al_{0.3}Ga_{0.7}As$/GaAs 2DEG-based heterostructure (the properties of the non-illuminated 2DEG are as follows: mobility $\mu \approx 10^6$ cm$^2$ V$^{-1}$ s$^{-1}$, density $n_s \approx 2.7 \times 10^{11}$ cm$^{-2}$, depth 100 nm) with standard split-gate techniques. It is anchored to a cold finger mechanically screwed to the mixing chamber of a dilution fridge with a base temperature of 70 mK. It is placed at the centre of the magnetic field produced by a solenoid. The coil allows to produce magnetic fields perpendicular to the 2DEG. The charge configuration of the triple-dot system is determined by measuring the conductance of the sensing dots biased with 300 μV; the current is measured using a current-to-voltage converter with a bandwidth of 10 kHz. The voltage on each gate can be varied on μs-timescales to allow exploration of the isolated configuration. Each green gate ($V_{M,1}$, $V_{M,2}$, $V_{M,3}$) in Fig. 1a is connected through a low temperature home-made bias-T to both DC and high bandwidth coaxial lines allowing gigahertz manipulations.

**Electron displacement procedure.** The voltage pulses to induce electron displacement are generated by an arbitrary waveform generator Tektronix 5014 C with a typical rise time (20–80%) approaching 0.9 ns. For $\tau_r$ equal to 1.7 ns, the pulse sequence presented in Fig. 3a is just reaching the programmed voltage amplitude. We can therefore assume that the electrons are only in the transfer phase during the displacement for $\tau_r = 1.7$ ns.

**Data availability.** All relevant data are available from the authors.

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

## Acknowledgements

We acknowledge technical support from the technological groups of the Institut Néel (Cryogenics, Electronics, Experimental engineering, Nanofab) as well as from Pierre Perrier and Henri Rodenas and valuable discussions with W. Coish and X. Hu. A.L. and A.D.W. acknowledge gratefully the support of the BMBF Q.com-H 16KIS0109, Mercur Pr-2013-0001 and the DFH/UFA CDFA-05-06. T.M. acknowledges financial support from ERC QSPINMOTION. Devices were fabricated at "Plateforme Technologique

Amont" de Grenoble, with the financial support of the "Nanosciences aux limites de la Nanoélectronique" Foundation and CNRS Renatech network.

## Author contributions

H.F. fabricated the sample and performed the experiments with the help of T.M. and C.B. H.F. and T.M. interpreted the data. H.F., C.B. and T.M. wrote the manuscript with the input of all the other authors. R.T. and P.-A.M. contributed to the experimental setup. A.L. and A.D.W. provided the high mobility heterostructures. All authors discussed the results extensively as well as the manuscript.

## Additional information

**Competing interests:** The authors declare no competing financial interests.

