## [Peer Review File · Nature Communications]

Reviewers' comments:

Reviewer #1 (Remarks to the Author):

The authors have employed a triple dot device in a ring configuration to compare coherent transport under static or rotation scenarios. They observe a longer spin mixing time when under rotation and interpret their results in terms of an additional effect analogous to that of motional narrowing in NMR. They are able to conclude that individual spins can be coherently displaced over a 5 micron distance. The ability to do this has potentially important consequences for Quantum information architectures using the spin quantum platform.

The device itself is non-trivial and the ability to tune it to the few electron regime controllably to be able to perform the various manipulations in this paper is in itself a significant achievement. The experiments are very impressive, rigorous and the conclusions important. The supplementary information contains some appropriate additional information.

My main criticism is related to the presentation in the paper. This is a novel device topology and so the reader requires a clear and detailed explanation of how the measurements are performed. I have found this lacking in places e.g. I still do not understand how figure 2(a) was obtained – in the text we are just told “the charge stability diagram is explored with a 50-ns voltage pulse on V1 and V2” and additionally the figure has arrows pointing in all directions on it. A clearer description of how this data was obtained should be given. Another example is mention of four additional lines in the text, presumably also in figure 2a also where only 2 are actually labeled. The explanation of figure 3(a) is confusing. The data is fascinating but what exactly is the point that is being made with the data in figure 3(a) about coherent transfer? How does the time constant deduced from the data (by multiplying the Number of turns by the time per turn) relate to T_2^* ?

The authors also have a tendency of making statements without any attempt at justification e.g. they mention that spin-orbit and transverse hyperfine are expected to become less efficient as the magnetic field is increased without any references. It is not obvious to me at least why the spin-orbit matrix element should be less efficient in this device as the field is increased. In fact the word “expected” is used 9 times in the paper. There is actually very little theory presented to back up their conclusions. They need to go through the paper and make it easier to follow for the general reader and justify their arguments in more detail.

The authors claim that that only classical spin transfer has been demonstrated. Arguably, this is not accurate e.g. Fujita et al. arXiv:1701.00815 and Busl et al. Nature Nanotechnology 8, 261 (2013) are two examples where coherent spin transfer is required.

In summary this is a very novel, interesting and impressive piece of work and is in my opinion suitable for publication in Nature Communications if the authors address the issues raised above.

Reviewer #2 (Remarks to the Author):

Review of the manuscript “Coherent long-distance displacement of individual electron spins” by H. Flentje et al., submitted for Nature Communications.

In this work, the authors address the very timely and very significant questions how quantum bits can be transferred in a future electron spin-based quantum computer. Far-distant transfer of qubit information is fundamental for most proposals of quantum computer architecture. So far, there were publications on single electron transport along a 1D channel and more recently on spin relaxation when displacing a single electron spin in an array of quantum dots. For the first time the authors experimentally investigate the spin coherence during single electron shuttling. More precisely, they use two electrons spins initialized in a singlet state, separated in two adjacent quantum dots and then displace them in a ring of three tunnel-coupled quantum dots. Finally, they test, whether the two electron spins are still in a singlet state by spin-state dependent electron tunnelling (spin-to-charge conversion) to an electron reservoir. This is not exactly the same as investigating the coherence during shuttling of a single electron spin shuttle, but it is an important

and novel experiment confirming an expected result for the material system GaAs/AlGaAs: Shuttling the electron spins reduced the impact of hyperfine noise due to motional narrowing. The analysis of the data is apart from the items mentioned below complete and the supplements add useful information. I invite the authors to revise the manuscript for some concerns about the data analysis and the readability of the paper before the final decision is reached.

The authors distinguish three regimes: (I) The two electrons are statically separated between two quantum dots. (II) The electrons are separated and tunnel from quantum dot to quantum dot (minimized transfer phase). (III) The electrons are separated and tunnel from quantum dot to quantum dot while the positions of the quantum dots themselves are non-static (minimized static phase).

For (III) in contrast to (II) pulse lengths shorter or equal to the pulse rise time are used. In regime (III) the longest spin coherence time is observed. It would increase the readability of the paper if the authors make clear what regimes they plot and compare in the figures. I will give some suggestions below.

Major points:

- Data recorded in regime (III) claimed to be single exponential, but the few data points recorded for a short separation time (Fig. 3a and Fig. 4) are not well fitted. The authors claim that these data (compared to II) cannot be fitted with a Gaussian in contrast to data recorded in regime (II). A comparison between a Gaussian and exponential fit will support the authors analysis. Alternatively, a reason for the discrepancy at short times should be given.
- Can the factor 8 increase of the coherence time comparing regime I and III be due to an increase of a factor 64 in the number of nuclei that electrons couple to in regime III? The authors just claim the "number of nuclei is drastically increased"
- The author claim that the tunnel-couplings of the device are tuneable up to GHz regime and they assume the tunnel couplings to be equal in a model. The authors should try to give measured numbers for the tunnel coupling. How equal is the tunnel coupling? How much is it changed by the external magnetic field applied in the out-of-plane direction of the 2DEG? In the supplements it is mentioned that some data is recorded in different tunnel-coupling regimes. This is quite typical for these kind of experiments, but should be mentioned in the main text. There seems to be a mistake as the authors claim that data in Fig 2 corresponds to the gate configuration of S5c (line 182 of the supplements), but clearly Fig 2a matches S5b.

Minor points:

- I suggest to swap panel 2d (static regime I) with panel 2b,c (introduction to non-static regimes II and III).
- I suggest to use different symbols for data recorded in regime I, II and III throughout the paper
- The data in the inset of Fig. 2e could be plotted in the main panel Fig 2e.
- At what magnetic fields is the data recorded in Fig. 2e (inset at 90 mT), orange dots at 150mT and blue at 200mT?
- It would be helpful to plot the borders of the stable charge regimes in Fig 2a. Fig 1b is not much helpful as the data is recorded sweeping voltages of different gates.

Response to the referees

We thank referee 1 for his/her careful reading of our manuscript, for his/her positive comments on our work and for recommending publication in Nature Communication. In the following we will answer the different questions raised in his/her report. We highlighted in green the modifications to the manuscript.

Reviewer 1:

- 1. I still do not understand how figure 2(a) was obtained – in the text we are just told “the charge stability diagram is explored with a 50-ns voltage pulse on V1 and V2” and additionally the figure has arrows pointing in all directions on it. A clearer description of how this data was obtained should be given.**

In the new manuscript we have improved the discussion of Fig. 2. and added a clearer description on how the data was obtained.

“Figure 2a presents where singlet-triplet spin mixing is occurring in the charge stability diagram. Each point is obtained by initialization to the singlet state at point R of Fig. 1b. Pulses of duration 50-ns and amplitude V_1 and V_2 are then simultaneously applied on the $V_{M,1}$ and $V_{M,2}$ gates respectively. The resulting two-electron spin state population for each (V_1, V_2) is then averaged over 150 single shot measurements.”

In Fig. 2a, the arrows represent the voltage pulses V_1 and V_2 to implement the separation of the electrons and their displacement in the triple-dot system. We changed the caption to clarify this point:

“The arrows represent the (V_1, V_2) voltage pulses used to implement the separation and the displacement of the electrons and presented in Fig. 2c.”

- 2. Another example is mention of four additional lines in the text, presumably also in figure 2a also where only 2 are actually labeled.**

We thank the reviewer for pointing this out. There are 4 lines of which 2 pairs each intersect at a charge degeneracy. To clarify the nature of the lines, additional arrows pointing at individual lines were introduced and now all 4 S-T+ crossings are pointed out in the figure. In addition, the approximate positions of the charge degeneracy lines in the figure now clearly delimit the respective charge configurations.

- 3. The explanation of figure 3(a) is confusing. The data is fascinating but what exactly is the point that is being made with the data in figure 3(a) about coherent transfer?**

Figure 3a plots data with τ_r as a function of the number of turns as described in the text. We therefore investigate how the spin dynamics is affected by the rotation speed of the displacement. As can be seen from the figure, the long decay constant of an exponential fit applied to the data is very much equal for all curves. This is a clear indication that the underlying process dominantly depends on the number of turns/distance travelled. This behavior is clearly unlike the previously observed behavior when spins are separated into two different dots where decay dominantly

depends on the separation time.

To address the point of the reviewer we rephrased this issue in the manuscript:

“Figure 3a shows the spin singlet probability as a function of N_t for different values of τ_r . The long decay is only dependent on N_t , and the underlying decay mechanism must therefore depend dominantly on the number of tunneling events or on the travelled distance of the electrons, rather than the separation time.”

4. How does the time constant deduced from the data (by multiplying the Number of turns by the time per turn) relate to $T2^*$?

As we assert in the manuscript, the loss of coherence of the two-electron singlet state is happening on a timescale that we called the spin coherence time. As was noticed in earlier investigations, once the electron are spatially separated, the exchange interaction becomes negligible and the good eigenstates are the Zeeman states (Up-Down and Down-Up). The decoherence processes of the singlet state then result in a complete mixing with the triplet states that we monitor in our experiment and give a measure of $T2^*$ for the singlet state. This is the timescale that we measured in our experiment with and without displacement.

With no displacement, the probability to detect the singlet state as a function of the separation time is characterized by a Gaussian decay with timescale $T2^*$.

With displacement, the separation time is actually equal to the multiplication of the number of turns N_t by the time per turn τ_r . We similarly observe the progressive decay of the singlet probability as a function of the separation time towards singlet-triplet mixing. It is no longer Gaussian but exponential. This is pointing at a change of the underlying physical process responsible for the singlet triplet mixing. The extracted timescale τ_{decay} is therefore the spin coherence time of the singlet state with displacement. This is the reason why we compare it to $T2^*$.

To clarify this point we add the following sentence in the paragraph “**Electron spin coherence with displacement**”:

“With displacement, the time τ_s where the electrons are separated and experience singlet-triplet mixing is equal to $3N_t\tau_r$.”

5. The authors also have a tendency of making statements without any attempt at justification e.g. they mention that spin-orbit and transverse hyperfine are expected to become less efficient as the magnetic field is increased without any references. It is not obvious to me at least why the spin-orbit matrix element should be less efficient in this device as the field is increased.

We clarified this specific point in the new manuscript and added references to justify our claims. The spin flip process (spin-orbit or lateral hyperfine stimulated by the motion) is expected to become ineffective for large magnetic fields. Indeed, the electron displacement results in an effective time-dependent magnetic field ΔB due to the change of the dot position. The smaller the change of the dot position, the smaller is ΔB . Due to the electron displacement in the moving dot, the amplitude of ΔB is expected to decrease with frequency. As only the ΔB component at the Larmor frequency

leads to spin-flips, the spin flip process become less and less effective for increasing magnetic fields. We added a reference related to lateral hyperfine interaction with motion (ref 27) and clarify this point in the manuscript:

“Moreover, individual spin-flip processes, stimulated by the electron motion and resulting from either spin-orbit or transverse hyperfine couplings, are then a possible decoherence channel for the electron spins^{27,28}. Due to the change of the dot position during the displacement procedure, both mechanisms are producing an effective time-dependent magnetic field ΔB on the displaced electrons leading eventually to a spin-flip. As a result, the two-electron system is able to leave the $S-T_0$ subspace. The smaller the change of the dot position, the smaller is ΔB . The displacement procedure in moving quantum dots implies therefore that the amplitude of ΔB decreases with frequency. As only the ΔB component at the Larmor frequency lead to spin-flips, the spin-flip processes are expected to become less and less efficient as the magnetic field is increased from 0 to 200 mT.”

6. In fact the word “expected” is used 9 times in the paper. There is actually very little theory presented to back up their conclusions. They need to go through the paper and make it easier to follow for the general reader and justify their arguments in more detail.

We have improved the manuscript to avoid repetition of the word “expected”. To our knowledge, the problem of spin decoherence due to motion in moving quantum dots has been investigated theoretically only very little. We are only aware of (ref 26) where this problem has been addressed. From this paper, we could come up with a qualitative understanding of our data. In the present manuscript, we have extended the discussion of the individual spin flip significantly to make it more accessible for the general reader audience. A quantitative comparison between the theory and the experiment is out of the scope of the paper. We are sure that our work will stimulate theorists.

“Moreover, individual spin-flip processes, stimulated by the electron motion and resulting from either spin-orbit or transverse hyperfine couplings, are then a possible decoherence channel for the electron spins^{27,28}. Due to the change of the dot position during the displacement procedure, both mechanisms are producing an effective time-dependent magnetic field ΔB on the displaced electrons leading eventually to a spin-flip. As a result, the two-electron system is able to leave the $S-T_0$ subspace. The smaller the change of the dot position, the smaller is ΔB . The displacement procedure in moving quantum dots therefore implies that the amplitude of ΔB decreases with frequency. As only the ΔB component at the Larmor frequency lead to spin-flips, the spin-flip processes are expected to become less and less efficient as the magnetic field is increased from 0 to 200 mT. These considerations are in agreement with the experimental findings of an enhancement of the spin mixing time and a gradual increase of the singlet population after 250 ns evolution with magnetic fields presented in Fig. 4. At 200 mT, the system remains, to a good approximation, in the $S-T_0$ subspace and the spin coherence time is expected to be limited by the residual longitudinal hyperfine interaction after motional narrowing²⁷. ”

7. The authours claim that that only classical spin transfer has been demonstrated. Arguably, this is not accurate e.g. Fujita et al. arXiv:1701.00815 and Busl et al. Nature Nanotechnology 8, 261

(2013) are two examples where coherent spin transfer is required.

To our opinion, the paper of Fujita et al. is the closest to our work. They operate a linear quadruple quantum dot system with two electrons and the idea is similar to ours to probe the coherence of the transfer, relying on the separation of the electrons. The coherent transfer is nevertheless limited to few hundreds of nanometers. Considering that this paper and our manuscript appeared almost simultaneously on Arxiv, we considered that classical spin transfer was not achieved when completing our work. We will discuss with the editor how to cite this work properly.

Concerning the paper of Busl et al, we think that the situation is very different to what we observe in our experiment. Indeed, the paper addresses the question of electronic transport properties through a triple dot system. Due to hybridization of the three dots, transport is observed through a superposition state with different occupation states. This is different from controlled coherent spin transfer as demonstrated in our manuscript. We have added this reference in connection with demonstration of the control over multidot systems.

Reviewer 2:

We thank referee 2 for his/her very careful reading of our manuscript, for his/her positive comments on our work, for his/her constructive suggestions and for recommending publication in Nature Communication. In the following we will answer the different questions raised in his/her report. We highlighted in green the modifications to the manuscript.

The authors distinguish three regimes: (I) The two electrons are statically separated between two quantum dots. (II) The electrons are separated and tunnel from quantum dot to quantum dot (minimized transfer phase). (III) The electrons are separated and tunnel from quantum dot to quantum dot while the positions of the quantum dots themselves are non-static (minimized static phase).

For (III) in contrast to (II) pulse lengths shorter or equal to the pulse rise time are used. In regime (III) the longest spin coherence time is observed. It would increase the readability of the paper if the authors make clear what regimes they plot and compare in the figures. I will give some suggestions below.

Major points:

- 1. Data recorded in regime (III) claimed to be single exponential, but the few data points recorded for a short separation time (Fig. 3a and Fig. 4) are not well fitted. The authors claim that these data (compared to II) cannot be fitted with a Gaussian in contrast to data recorded in regime (II). A comparison between a Gaussian and exponential fit will support the authors analysis. Alternatively, a reason for the discrepancy at short times should be given.**

The data agrees best with an exponential fit. We appended a new section (section VII) in the supplementary materials showing a comparison of a best Gaussian fit and a best exponential fit on the data presented in Fig.2e (the same as for Fig.4 at 200 mT). From this new figure one can clearly see that the Gaussian model fits less well the data.

In Fig3a, the observed discrepancies at short times are explained by the double decay explained in

the context of the inset of Fig2e and are the results of the “static” phase. For Fig4, the discrepancies at short time are less clear on the data than in Fig3a.

We added a reference to the supplementary material section VII in the main text where a comparison between the exponential and Gaussian fit is realized on the the data of Fig2e.

We added a comment in the caption of Fig3a to discuss the behavior at short times:

“The fast decay associated to the static phase is observed for small number of turns. The data after the static phase are fitted with an exponential decay with a characteristic number of turns N_{Decay} .”

2. Can the factor 8 increase of the coherence time comparing regime I and III be due to an increase of a factor 64 in the number of nuclei that electrons couple to in regime III? The authors just claim the “number of nuclei is drastically increased”

In the case of longitudinal hyperfine-limited coherence and static dots, it is true that the factor of increased nuclear spins should be related to the square of the factor of increased coherence time. At least in the case of 200 mT where spin-flip processes are suppressed, it will then be tempting to associate the increase of the coherence time with an increase of the number of nuclei coupled to the electrons during displacement.

However, the area visited by the electrons is only increased by a factor of 15 with motion (assuming a dot area of 20 nm x 20 nm and a path length of 330 nm) which is quite different with the factor 64 expected from the coherence time. Three comments can be made at this point: first, the assumption that the two distributions of effective magnetic field experienced by the two electrons via hyperfine interaction are different is not true in our experiment since the electrons are displaced on the same path. We speculate that it could result in a further increase of the coherence time. Second the displacement of the electrons can be different from a straight line and it can result also in an increase of the area explored during the electron motion and of the coherence time. Finally, the two compared situations are different from a conceptual point of view: a large dot where the electron interacts with 64 times more nuclei at the same time and an electron in a moving quantum dot where it interacts sequentially with a larger number of nuclei are not expected to give exactly the same spin coherence time.

In conclusion, the quantitative explanation of the increase by a factor 8 is not entirely clear at present and due to the upper limit set by the other decay process it is unfortunately not possible to give a quantitative estimate of the exact number of nuclei.

The question is in our opinion very interesting and further experimental as well as theoretical studies are needed to answer this question.

We have therefore opted for a qualitative formulation of the increase of the number of nuclei coupled to the electrons:

“As a consequence, the number of nuclei coupled to the electrons is drastically increased and a process similar to motional narrowing observed in liquid nuclear magnetic resonance experiments^{25,26} is expected to result in an increase of the electron spin coherence.”

3. The author claim that the tunnel-couplings of the device are tunable up to GHz regime and they assume the tunnel couplings to be equal in a model. The authors should try to give measured numbers for the tunnel coupling. How equal is the tunnel coupling? How much is it changed by the external magnetic field applied in the out-of-plane direction of the 2DEG?

In this experiment, we could tune the coupling from kHz (data not shown) to the situation presented in the manuscript where coupling are close to few GHz. Concerning the magnetic field range used in the experiment, we do not expect a drastic change of the size of the dot which would affect the tunnel couplings. Indeed, the coupling is dominated by the gate confinement in this magnetic field regime, ($\hbar\omega_0 \sim 1\text{meV} \gg 1/4 \hbar\omega_c \sim 0.1\text{meV}$). During the experiment, tuning the tunnel coupling was done by changing the gate voltages as shown in Fig S5.

Direct measurement of the tunnel coupling is rather difficult in the range of GHz. We used both the charge stability diagrams and the spin map to guide the tuning. An indirect proof of the coupling was obtained from the spin map (see Figure 2a and S5). In particular, the observed separation between the S-T+ lines and the mixing regions are an indication that the coupling is on the order of several GHz and the transfer of charge is adiabatic. The observed separation between the mixing regions is pretty much the same (see fig 2a) which is an indication that the couplings are similar.

It is worth noting that the enhanced spin coherence for displaced electrons that we report is not very sensitive to a change in the tunnel coupling (see Supplementary section VI).

We have added a sentence in the caption of Fig2a describing the spin-mixing map to point out that additional data are shown in the Suppl. Mat. :

“The influence of the tunnel-coupling on the spin-mixing map is discussed in Supplementary Section V.”

In the Supplementary Section V, we added a sentence concerning the magnetic field dependence of the tunnel-coupling and the symmetry of the tunnel-coupling:

“In the main text, the presented results have been mostly obtained for a specific tunnel-coupling between the dots. In this section, the spin coherence time after displacement for different tunnel-couplings between the dots is analysed. The strength of the tunnel-couplings can be changed by controlling the potential of the red gates in Fig. 1a. It is worth noting that the tunnel coupling is not affected by the perpendicular magnetic field in the range (0-200mT) used in the experiment. The tunnel coupling can directly be witnessed on the spin mixing maps in Fig S5b-d: for decreasing tunnel-coupling, the separation between the three mixing regions is progressively vanishing. We observe that the exchange interaction at the single-electron crossing between two dots when the electrons are separated is reduced and becomes negligible in comparison with the hyperfine interaction. Moreover, the separation in gate voltage space between the S-T₀ mixing region and the S-T+ crossing lines is progressively reduced until it vanishes completely. Such observations are in agreement with a progressive reduction of the singlet level repulsion induced by the tunneling process and are consistent with a reduction of the tunnel-couplings between the dots. Moreover, the observed separation between the S-T₀ mixing regions is pretty much the same (see Fig. S5b) which is an indication that the tunnel-couplings between the dots are similar.”

- 4. In the supplements it is mentioned that some data is recorded in different tunnel-coupling regimes. This is quiet typical for these kind of experiments, but should be mentioned in the main text. There seems to be a mistake as the authors claim that data in Fig 2 corresponds to the gate configuration of S5c (line 182 of the supplements), but clearly Fig 2a matches S5b.**

We presented a typical spin mixing map in Fig. 2a. It is not the one associated with the displacement data presented in Fig2 and Fig4. We chose it as it was the more illustrative of the underlying physics

with clearly separated mixing regions. As explained in the supplementary materials, we did not notice any significant change of the mixing process by changing the tunneling. The displacement data of Fig2-Fig4 and Fig3 are associated to different tunings as explained in the supplementary materials. We changed the caption of Fig2, Fig3 and Fig4 to clarify the tunnel-coupling conditions for each set of data.

Caption Fig3 at the end of paragraph (a.): All the data presented in this figure are in a tunnelling condition slightly different than Fig. 2 and 4 (see Supplementary Section V).

Caption Fig2 at the end of paragraph (a.) : The influence of the tunnel-coupling on the spin-mixing map is discussed in Supplementary Section V.

Caption Fig2 at the end of paragraph (e.) : We notice very similar spin mixing time-evolution with displacement for different tunnel-coupling conditions (see Supplementary section V).

Minor points:

- 5. I suggest to swap panel 2d (static regime I) with panel 2b,c (introduction to non-static regimes II and III).**

We thank the reviewer for this suggestion and have adopted this change in the revised manuscript.

- 6. I suggest to use different symbols for data recorded in regime I, II and III throughout the paper**

This labelling is according to us only valid for Fig2e Fig3a and Fig4. There is only one instance for the regime (II) (Fig. 3b). The reviewer's definition of regime (I) is plotted in Fig. 2b, and correspond to the reference of decoherence without displacement. We propose to add in the caption of Fig2e Fig3a Fig4 a sentence connecting the figure to the transfer regime.

Caption Fig2e: "e, Singlet probability as a function of the time τ_s , for the case where the electrons are rotating between separated charge configurations with $\tau_r = 1.7$ ns (electron displacement with minimized "static" phase)."

Caption Fig3a: "b, Singlet probability as a function τ_s for N_t equal to one and obtained by increasing τ_r (electron displacement with minimized "transfer" phase)."

Caption Fig4: "Singlet probability as a function of the time τ_s for the case where the electrons are rotating in the triple-dot system for different B at fixed $\tau_r = 1.7$ ns (electron displacement with minimized "static" phase)."

- 7. The data in the inset of Fig. 2e could be plotted in the main panel Fig 2e.**

We had considered this in an early draft of the manuscript, however, the additional cluttering takes away from the data in terms of clarity. Note that the data in the inset of Fig. 2e is at a different magnetic field and at a different rotation speed and was only considered to constitute experimental indication for a two-phase behavior.

The case of differing rotation speeds is more systematically explored later in Fig. 3a, where the appearance of two decay times becomes more pronounced with reduced rotation speeds. We therefore consider that the data plotted in the inset of Fig. 2e is of lower importance compared to the data of Fig. 2e and would consider the present form optimal.

We have removed the data without displacement on Fig2e and indicated the magnetic field on the inset of Fig2e.

8. At what magnetic fields is the data recorded in Fig. 2e (inset at 90 mT), orange dots at 150mT and blue at 200mT?

We agree with the referee that indication of the magnetic field values in Fig 2 improves the readability of the figure.

We have removed the data without displacement on Fig2e and indicated the magnetic field on the inset of Fig2e.

9. It would be helpful to plot the borders of the stable charge regimes in Fig 2a. Fig 1b is not much helpful as the data is recorded sweeping voltages of different gates.

We want to keep Fig1b as it shows on a large gate span the charge response and the control of the triple dot system with two electrons. It is the first time that such a graph has been obtained in the isolated configuration.

To address this point of the reviewer concerning the Fig 2a and make the figure clearer to the reader, we added the estimated positions of the charge degeneracy lines in Fig. 2a with dashed lines.

REVIEWERS' COMMENTS:

Reviewer #1 (Remarks to the Author):

The authors have improved the manuscript and I now recommend the paper be published in Nature communications. This is really a very beautiful piece of work that will be of great interest to the quantum dot community. I congratulate the authors.

One final point I found figure 2(d) confusing. I don't understand the orange numbers- why are they there? Also the inclusion of "Nt" in the figure 2C is not necessary and a bit confusing.

Reviewer #2 (Remarks to the Author):

Review of the revised manuscript "Coherent long-distance displacement of individual electron spins" by H. Flentje et al., submitted for Nature Communications.

The authors well addressed all points raised by the referees. The data analysis and readability of the paper improved considerably. Now the manuscript should be accepted for the publication in Nature Communications.